# An Overview of Obesity, Cholesterol, and Systemic Inflammation in Preeclampsia

**DOI:** 10.3390/nu14102087

**Published:** 2022-05-17

**Authors:** Morgan C. Alston, Leanne M. Redman, Jennifer L. Sones

**Affiliations:** 1Departments of Veterinary Clinical Sciences, School of Veterinary Medicine, Louisiana State University, Baton Rouge, LA 70803, USA; malsto4@lsu.edu; 2Reproductive Endocrinology and Women’s Health Laboratory, Pennington Biomedical Research Center, Baton Rouge, LA 70808, USA; leanne.redman@pbrc.edu

**Keywords:** preeclampsia, inflammation, hypertension, obesity, cytokines, metabolic abnormalities, leptin, cholesterol

## Abstract

Preeclampsia (PE), an inflammatory state during pregnancy, is a significant cause of maternal and fetal morbidity and mortality. Adverse outcomes associated with PE include hypertension, proteinuria, uterine/placental abnormalities, fetal growth restriction, and pre-term birth. Women with obesity have an increased risk of developing PE likely due to impaired placental development from altered metabolic homeostasis. Inflammatory cytokines from maternal adipose tissue and circulating cholesterol have been linked to systemic inflammation, hypertension, and other adverse outcomes associated with PE. This review will summarize the current knowledge on the role of nutrients, obesity, and cholesterol signaling in PE with an emphasis on findings from preclinical models.

## 1. Maternal Risk Factors of PE 

Preeclampsia (PE) is a leading cause of maternal and fetal mortality and morbidity [1]. Worldwide, 5–7% of all pregnancies are affected, resulting in 70,000 maternal deaths and 500,000 fetal deaths from PE each year [2]. In human pregnancy, the clinical presentation of PE includes new-onset hypertension along with the development of proteinuria, endothelial dysfunction, or another accompanying signs/symptoms of multiorgan dysfunction [3]. Maternal hypertension is determined when two blood pressure readings taken at least 6 h apart both display a systolic value greater than 140 mmHg and a diastolic value greater than 90 mmHg [4]. Signs and symptoms present after 20 weeks of gestation [4]. Prior to 34 weeks of gestation, PE is considered early onset; however, the disorder can arise through to delivery, and after 34 weeks is considered late onset [4]. Many factors are suspected to contribute to the development of PE, with family history, advanced maternal age in pregnancy (>35 years of age), and pre-existing medical conditions (e.g., obesity, hypertension, diabetes or renal dysfunction) conferring the highest risk [3]. This review specifically describes the potential mechanism linking obesity, cholesterol, and systemic inflammation to PE. 

PE is associated with uterine and placental abnormalities and adverse fetal co-morbidities such as intrauterine growth restriction and preterm birth [3]. In a healthy pregnancy, placentation begins with trophoblast invasion into the uterine arteries. Arterioles are remodeled into high capacitance and high flow vessels to provide the fetus with nutrients [2]. In preeclamptic pregnancy, trophoblast invasion is inadequate and thus vascular remodeling is impaired, which results in narrowed maternal vessels and placental ischemia [2]. Poor perfusion of oxygen within the uterine vasculature can lead to hypoxia, oxidative stress and the release of antiangiogenic factors such as soluble fms-like tyrosine kinase 1 (sFLT1) [2]. sFLT1 increases vasoconstriction and contributes to maternal metabolic dysfunction by raising blood pressure [5]. An increase in sFLT1 is also associated with a decrease in placental growth factor (PIGF), a member of the vascular endothelial growth factor (VEGF) family [5]. The imbalance of sFLT1 and PIGF further contributes to the development of placental dysfunction and the clinical presentation of PE [5]. 

## 2. Obesity and Systemic Inflammation in Pregnancy 

Obesity affects more than 2 billion people worldwide, thus explaining the term “globesity” [6]. Compared to normal weight, obesity causes inflammatory changes in the adipose tissue that increase the risk for PE by worsening metabolic abnormalities such as insulin resistance, inflammation, and atherosclerotic disease [7]. With the incidence of maternal obesity and cases of PE rising worldwide, there is an urgent need to understand whether and how the attenuation of maternal obesity can improve PE and its downstream outcomes [8]. 

Adipose tissue, composed of adipocytes, immune cells, and stromal vascular cells, acts as an endocrine organ that participates in modulating systemic inflammatory and immune responses [9]. Central or visceral adipose tissue depots release adipokines such as tumor necrosis factor alpha (TNF-α) and interleukin-6 (IL-6) from adipose tissue macrophages and other immune cells that promote cell-to-cell signaling of local and systemic inflammation [10]. For pregnancy success, the maternal-fetal interface undergoes significant changes with trophoblast invasion and remodeling of decidual vasculature [11]. In pregnancies affected by obesity, it is hypothesized that cross-talk between maternal adipose tissue and the maternal-fetal interface may contribute to improper vascularization of the placenta due to high circulating proinflammatory immune cells [11,12]. The release of proinflammatory cytokines (TNF-α and IL-6) and other anti-angiogenic factors from both the adipose tissue and ischemic placenta can result in maternal hypertension and fetal growth restriction [12]. 

## 3. Maternal Obesity and Metabolic Abnormalities 

Often women with obesity demonstrate metabolic abnormalities such as increased circulating leptin, glucose, insulin, and cholesterol, that play a cumulative and vital role in the development of PE [13]. Leptin is a pleiotropic hormone that is involved in energy homeostasis by regulating satiety in the brain, temperature regulation, and glucose metabolism [14]. Leptin levels increase normally with eating; however, levels increase beyond a normal range in obesity [15]. Additionally, leptin is a cytokine with both innate and adaptive immune responses that specifically promote inflammatory conditions [16]. Hyperleptinemia in people with obesity has been associated with a low-grade inflammatory state that is responsible for the development of autoimmune disease as well as reproductive and gestational disorders [15,16]. The two main sources of leptin during pregnancy are the adipose tissue and the placenta [15]. In normal pregnancy, leptin regulates vascular function and systemic inflammatory responses that impact placentation [15]. 

Cholesterol is also thought to contribute to pregnancy outcomes [7]. Circulating cholesterol is derived from two sources: dietary intake or cellular biosynthesis. Cellular cholesterol biosynthesis and cholesterol homeostasis is maintained primarily by the liver [17], but cholesterol is also stored in adipose tissue [18]. Recent studies suggest that adipose tissue serves as a major storage site for free cholesterol and an increase in cholesterol accumulation parallels obesity [18]. In an obese state, approximately 50% of the body’s cholesterol is stored in adipose tissue [18]. Cholesterol serves two important functions: a structural role providing cellular support at the level of the plasma membrane and a functional role with cell signaling [19]. The ratio of cholesterol carrying molecules, low density lipoproteins (LDLs), and high-density lipoproteins (HDLs) are also important in the atherogenic process. Increased levels of LDLs, molecules that transport cholesterol from the liver to peripheral tissue, can lead to plaque build-up, atherosclerotic disease, and downstream systemic inflammation [20]. Cholesterol has also been shown to cause an increase in adipose tissue inflammation and adipose tissue remodeling [18]. 

During pregnancy, an increase in maternal cholesterol, which is often termed maternal physiological hypercholesterolemia (MPH), is a normal physiological adaptation to ensure proper fetal development [21]. However, higher than normal cholesterol during pregnancy, termed maternal supraphysiological hypercholesterolemia (MSPH), is associated with endothelial dysfunction and atherosclerotic lesions in the placental vasculature [22]. Normal endothelial and macrophage function are reliant on the proportion of HDLs and LDLs for differentiation [21]. Macrophages adopt pro- or anti- inflammatory functions depending on the ratio between HDLs and LDLs; higher circulating levels of LDLs and lower levels of HDLs are reflective of processes promoting atherogenesis [21]. Taken together, maternal adiposity and dyslipidemia may play a causal role in PE. 

## 4. Mechanistic Insights from Human Pregnancy Studies

In the past 20 years, several clinical studies supporting the maternal dyslipidemia and obesity hypotheses of PE have been published [12,23,24,25,26,27,28]. Specifically, an abnormal lipid profile in pregnancy has shown to be positively correlated with the development of maternal atherosclerosis and endothelial dysfunction [23]. Moreover, elevated levels of triglycerides, total cholesterol, and LDL cholesterol paired with low levels of HDL are characteristic of PE pregnancies [25]. Table 1 summarizes compiled data from cross-sectional studies as well as meta-analysis from human studies to further explain the contribution of an abnormal lipid profile in the development of pregnancy disorders such as PE. Studies in Table 1 compare serum lipid measurements in women with PE to control women with healthy pregnancies throughout different stages of gestation (1st, 2nd, and 3rd trimesters). These studies report the correlation between elevated serum lipid levels to the development and risk of PE. Additionally, some studies test the effectiveness of nutrient treatments in reversing the adverse outcomes associated with PE. 

## 5. Mechanistic Insights from Animal Models of PE 

In vitro studies that investigate the intersection of maternal obesity and circulating lipids on placental development are lacking. Advancements in technology will one day provide model systems such as organoids to study PE [29]; however, the technology has not yet been developed to uncover the mechanistic complexities of PE. Thus, animal models serve as a potential way to test hypotheses that may prevent the adverse outcomes seen in PE, and results can be used to translationally improve PE outcomes seen in high-risk pregnant women with obesity. The Blood Pressure High (BPH)/5 mouse model serves as a preclinical model for PE studies. The BPH/5 female mouse has pre-existing hypertension that increases spontaneously in pregnancy and other adverse outcomes of PE observed in human pregnancy such as proteinuria, endothelial dysfunction, and fetal growth restriction [8]. Compared to controls, pregnant BPH/5 female mice are hyperphagic with increased white adipose tissue (WAT) mass [8] that overexpresses proinflammatory cytokines (TNF- α and IL-6). They are also dyslipidemic with increased circulating cholesterol and leptin [30].

The reduced uteroplacental perfusion pressure (RUPP) rat displays PE associated signs/symptoms such as hypertensive oxidative stress, endothelial dysfunction, and fetal growth restriction [31]. Feeding the RUPP rat a high cholesterol diet beginning in early gestation results in a severe model of PE with signs of maternal neurological disturbances [32]. The high cholesterol diet results in elevated levels of cholesterol that increase oxidative stress and thus impair normal endothelial and vascular functioning specifically in the brain [32,33]. 

Studies in pregnant rodents have elegantly demonstrated the relationship between leptin and other atherogenic factors that are thought to promote inflammation, ischemia, and endothelial dysfunction, all of which are key features of PE [34]. The subcutaneous administration of leptin to pregnant rats resulted in adverse outcomes such as increased blood pressure, dyslipidemia, endothelial cell dysfunction, and overall increased markers of systemic inflammation [34]. 

Likewise, the lectin-like oxidized low-density-lipoprotein receptor-2 (LOX1) overexpressing mouse is a model of vascular dysfunction as increased LOX1 induces angiotensin activation and leads to PE associated outcomes [35]. LOX1 has been associated with the development of hypertension, insulin resistance, hyperlipidemia and complications associated with obesity [36]. 

Research studies have also used mouse models to demonstrate the importance of heme-oxygenase 1 (HO-1), an enzyme expressed in the placenta [37]. HO-1 has protective functions in the placenta and contributes to the maintenance of proper maternal vascularization [37]. Human patients with pregnancy disorders have commonly showed a down-regulation of HO-1 expression in their placentas; thus, lower than normal levels of HO-1 has been correlated to the development of pregnancy disorders such as PE [37]. Breeding mice heterozygous for HO-1 (HO-1+/−) resulted in decreased litter sizes and pup weights compared to wild type (WT) breeding [37]. Animal models such as BPH/5 mice, the RUPP rat, and HO-1 transgenic mice among others serve as a practical preclinical system to test improvements that could potentially mitigate adverse outcomes seen in women who are at high risk for developing PE. 

## 6. Insights from Pharmacological and Lifestyle Interventions in Pregnant Women and Rodents 

The American College of Obstetricians and Gynecologists suggests that women who are high risk for developing PE (extensive family history of PE, advanced maternal age, or pre-existing medical conditions such as obesity, diabetes, or renal dysfunction) take a low dose of aspirin (81 mg/d) starting after 12 weeks of gestation [38]. Aspirin acts to normalize angiogenic imbalance by normalizing elevated circulating sFLT1 in a preeclamptic placenta [39]. Human studies have shown that low-dose aspirin (81 mg/day) given from six to 36 weeks of gestation did not significantly increase the risk of emergency medical visits or potential side effects such as nausea, vomiting, rash, vaginal bleeding, diarrhea, or others [40,41,42]. 

Additionally, celecoxib, a COX-2 inhibitor, has also been shown to restore angiogenic imbalance in the placenta in addition to attenuating maternal hypertension and intrauterine fetal growth restriction in the BPH/5 mouse model [43]. It has been shown that the elevated levels of COX-2 at implantation contribute to factors that can negatively impact decidualization and placentation; however, levels of angiogenic factors are normalized after celecoxib administration [43]. Reports of human studies have shown that high doses of celecoxib administered late in gestation can have adverse fetal effects such as premature closure of the ductus arteriosus [44]. However, some studies have begun to show that administration in pregnancy can impede preterm labor without having detrimental effects on the fetus [45]. 

Other pharmacological interventions such as the use of statins have shown the potential to improve and counteract the PE phenotype [46]. Statin administered between 12–16 weeks of pregnancy and continued until delivery stimulates trophoblast invasion, improves placental blood flow, and provokes anti-inflammatory agents, which all work to diminish the adverse effects associated with PE [46]. Although there has been promising data on the use of statins in pregnancy, few human studies have successfully shown that statin administration is safe in pregnancy [47,48]. Statin is most commonly deemed as teratogenic and hence contraindicated in pregnancy due to increased risk of congenital abnormalities as well as other complications [49]. 

Tadalafil, a long-acting phosphodiesterase 5 (PDE5) inhibitor, is another pharmacotherapy under evaluation for the attenuation of PE [50]. In the RUPP model of PE, Tadalafil successfully improved maternal hypertension and attenuated the PE phenotype [50]. Human studies have shown that tadalafil has not adversely affected cardiac function in pregnant women [51]; however, few studies have reported the safety profile for both mother and fetus [52]. Sildenafil citrate, a similar inhibitor, has been investigated in a rodent model for its ability to improve blood flow in the uterine artery and for its vasodilatory effects [53]. Limited studies have shown that sildenafil does not contribute to adverse maternal or fetal side effects [54]. More studies are needed to prove the safety of tadalafil and sildenafil administration in pregnancy. 

Unfavorable pregnancy outcomes seen with pharmacological therapy have emphasized the need for lifestyle intervention to attenuate maternal adipose tissue and abnormal maternal metabolic health. Published studies have demonstrated the positive effects of lifestyle interventions, specifically adipose tissue reduction by reduced food intake or increased activity, in pregnancy for both animal models and women.

In the female, hyperphagic BPH/5 mouse, pair feeding to lean control mice (equivalent to 25% calorie restriction) reduced white adipose tissue mass around the reproductive organs [30]. Pair feeding also decreased circulating leptin and WAT-derived inflammatory cytokines, but had no effect on serum cholesterol [30]. Molecular analyses showed pair feeding BPH/5 in early pregnancy decreased prostaglandin synthase 2, the enzyme responsible for prostaglandin formation, in the reproductive WAT and the implantation sites [30]. These molecular results preceded adverse late gestational PE outcomes in BPH/5, i.e., maternal increases in blood pressure. Another study using a mouse model of enriched environment that induced weight loss, displayed the beneficial metabolic improvements with attenuation of maternal obesity in both the pregnant dam and offspring [55]. For example, maternal wight loss in pregnant female mice improved glucose tolerance and insulin sensitivity as well as decreasing adipose tissue [55]. Similarly, offspring whose mothers had undergone maternal weight loss demonstrated similar metabolic improvements as well as a down-regulation of numerous lipid and cholesterol biosynthesis genes [55]. 

In human pregnancy, gestational weight loss resulted in a decreased incidence of PE, non-elective cesarean, preterm delivery, small/large gestational birth weight, and perinatal mortality [56]. Together, these studies in both animal and human models demonstrate the potential beneficial effects of maternal weight loss and adipose tissue reduction on PE and its associated adverse pregnancy outcomes. 

## 7. Conclusions

PE is a disorder of pregnancy characterized by maternal hypertension and fetal growth restriction [3]. PE is thought to be caused by improper placental vascularization and subsequent increased systemic inflammation as the maternal signs of PE resolve after delivery of the placenta [3]. PE continues to be a leading cause of maternal and fetal morbidity and mortality with limited knowledge regarding the prevention and attenuation of onset or worsening of symptoms [1]. Inflammatory adipokines and cytokines from maternal adipose tissue and circulating cholesterol have been linked to the development of systemic inflammation, pregnancy-induced hypertension, an imbalance of placental angiogenic factors, and other adverse PE outcomes [10,20,43]. Therefore, the reduction of adipose tissue and associated adipokines is hypothesized to be associated with adequate placental angiogenesis and the overall improvement of adverse outcomes associated with PE by altering inflammatory responses [30]. Additionally adipose tissue serves as a major reservoir for free cholesterol [18]. Supraphysiological levels of cholesterol in pregnancy are hypothesized to contribute to the improper vascularization of the placenta and overall systemic inflammation seen with PE [22]. Rodent studies have shown that increased dietary cholesterol leads to an increased accumulation of differentiated pro-inflammatory adipose tissue macrophages [7]. In summary, the adipose tissue plays an important role in pregnancy. Increased maternal adipose tissue mass and cholesterol promote metabolic abnormalities and systemic inflammation, which may contribute to improper placental vascularization that leads to PE with increased maternal blood pressure and fetal growth restriction (Figure 1). Future studies are needed to further elucidate the contribution of pharmacotherapy as well as lifestyle interventions, specifically maternal weight loss, and PE outcomes in high-risk women with obesity. 

## Figures and Tables

**Figure 1 nutrients-14-02087-f001:**
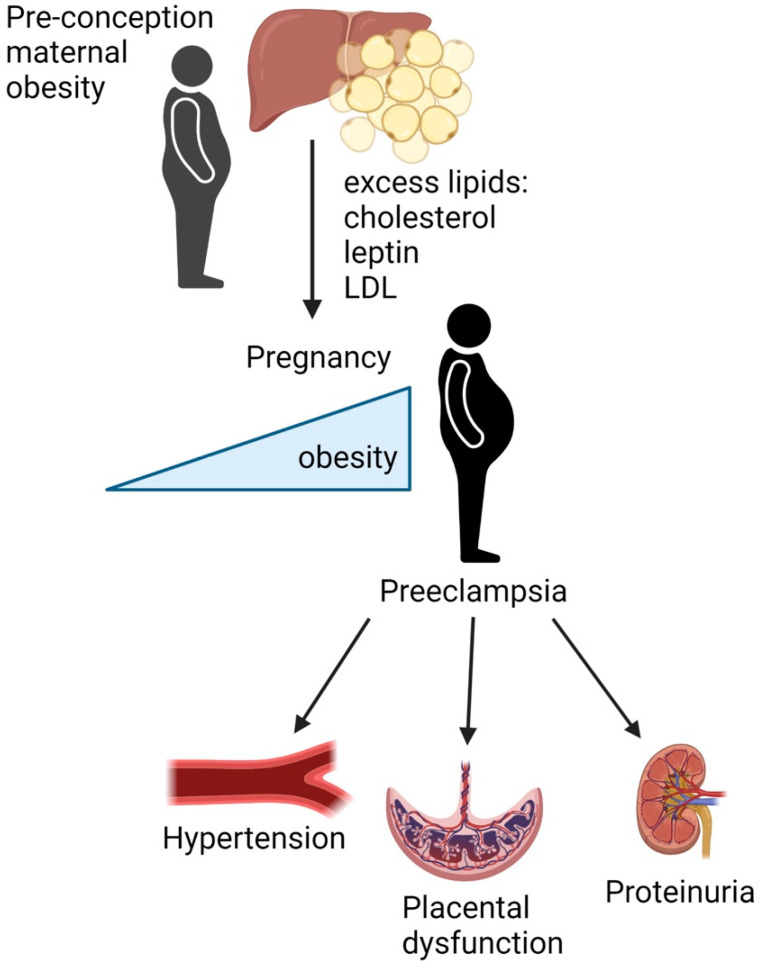
Maternal obesity and dyslipidemia promotes preeclampsia through abnormal placental development. The liver oxidizes lipids and excess lipids are shunted to adipose tissue for storage. As adipocytes become hypertrophic in obesity, there is an increase in leptin, total cholesterol, as well as low-density lipoproteins (LDL) in the maternal circulation. This increase in lipid accumulation and dyslipidemia results in endothelial dysfunction and an overall increase in inflammation with increased weight gain during pregnancy, which may contribute to improper vascularization of the placenta and adverse outcomes associated with preeclampsia: elevated maternal blood pressure and fetal growth restriction. Made with BioRender.com (accessed on 11 May 2022).

**Table 1 nutrients-14-02087-t001:** Human studies providing evidence for a link between dyslipidemia and preeclampsia.

Subjects	Gestational Age	Measures	Relationship to PE	Treatments/ Nutrients	Citation
six with early onset PE (pilot study, no control)	24–27 weeks	Cholesterol LDL Apo B	Lipoprotein remnants = endothelial dysfunction	Apheresis (reduced ApoB)	Contini et al., 2018 [24]
7369–1975 with PE and 5394 healthy (meta-analysis)	first and second trimesters third trimester	Cholesterol LDL TG Cholesterol TG HDL	Elevated in PE Elevated in PE Elevated in PE Elevated in PE Elevated in PE Low in PE	--	Spracklen et al., 2014 [25]
105 PW- 50 treatment, 55 placebo 60 healthy controls	second trimester	Oxidized low density lipoproteins (oLAB)	oLABs contribute to intrauterine growth retardation	Chokeberry Anthocyanins (controls oxidative stress)	Pawlowicz et al., 2000 [26]
100 PW–50 treated, 50 placebo	third trimester	Cholesterol LDL HDL TG	--	Garlic Tablet (reduces hypertension)	Ziaei et al., 2001 [27]
173 with PE 186 healthy controls	Post-partum	HDL Triglycerides	Higher levels, decreased risk of PE Higher levels, increased risk of PE	--	Williams et al., 2003 [28]

Abbreviations: Preeclampsia (PE), Pregnant Women (PW), Low Density Lipoprotein (LDL), Apolipoprotein B (ApoB), High Density Lipoprotein (HDL), Triglycerides (TG).

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
