# Peer review of "An Overview of Obesity, Cholesterol, and Systemic Inflammation in Preeclampsia"

_nutrients, 2022, doi:10.3390/nu14102087_

Round 1
Reviewer 1 Report
The authors have submitted a very well written review of the links between obesity, systemic inflammation and preeclampsia. I like this article overall but have a few suggestions the authors may want to consider.
1) It would be good if the authors added a few sentences describing the link between the development of the placental vasculature and preeclampsia. It comes up once or twice when the author discusses potential mechanisms so it would be good to introduce the idea early on. I also feel like the authors could focus a little more on the role of damage or under development of the placental vasculature plays in the disease.
2) The section on insight from animal models seems a little under cooked. The authors should consider moving the BPH/5 and RUPP models (2nd and 3rd paragraph of this section) to be the 1st and 2nd paragraphs. The short paragraphs about Leptin and Lox1 may be combined since they presumably work through similar mechanisms. There is quite of bit of work discussing the link between leptin and inflammation that should be discussed in a little more detail. It may also be worth including some information about heme-oxygenase 1 (HO-1) mice? Finally, the last sentence just seems to be hanging there. Maybe the authors can move it to the beginning of the section to introduce the idea of using animal models to find new pathways worthy of study?
3) The insights from pharmacological and lifestyle interventions section could also use a little tightening up. Is it possible to combine the portions discussing aspirin and celecoxib since they are both Cox inhibitors? Maybe use the space that frees up to discuss how these drugs "normalize angiogenic imbalance" in a little more detail. The authors briefly mention sFLT1 but do not describe its significance.
Author Response
The authors have submitted a very well written review of the links between obesity, systemic inflammation and preeclampsia. I like this article overall but have a few suggestions the authors may want to consider.
- It would be good if the authors added a few sentences describing the link between the development of the placental vasculature and preeclampsia. It comes up once or twice when the author discusses potential mechanisms so it would be good to introduce the idea early on. I also feel like the authors could focus a little more on the role of damage or under development of the placental vasculature plays in the disease.
We thank the reviewer for pointing out this deficiency. We have added references providing mechanistic details of placental vasculature development and how it is perturbed in preeclampsia to the second paragraph. To both address this first comment as well as the third below, sFlt was introduced earlier in the manuscript to provide context for the “Insights from Pharmacological and Lifestyle Interventions” section.
- The section on insight from animal models seems a little under cooked. The authors should consider moving the BPH/5 and RUPP models (2nd and 3rd paragraph of this section) to be the 1st and 2nd paragraphs. The short paragraphs about Leptin and Lox1 may be combined since they presumably work through similar mechanisms. There is quite of bit of work discussing the link between leptin and inflammation that should be discussed in a little more detail. It may also be worth including some information about heme-oxygenase 1 (HO-1) mice? Finally, the last sentence just seems to be hanging there. Maybe the authors can move it to the beginning of the section to introduce the idea of using animal models to find new pathways worthy of study?
The authors respectfully agree with your comments on the animal model section. We have moved the order of BPH/5 and RUPP, combined the leptin, which was beefed up, and Lox 1 sections as suggested as well as added a paragraph of Hmox1 transgenic mice.
- The insights from pharmacological and lifestyle interventions section could also use a little tightening up. Is it possible to combine the portions discussing aspirin and celecoxib since they are both Cox inhibitors? Maybe use the space that frees up to discuss how these drugs "normalize angiogenic imbalance" in a little more detail. The authors briefly mention sFLT1 but do not describe its significance.
Thank you for your suggestions here. We have combined aspirin and celecoxib points and provided relevance to sFlt and angiogenic balance in an effort to make the connection between these therapeutics and improved placental vasculature development in pregnancy which is key to the pathogenesis of preeclampsia.
Reviewer 2 Report
The work entitled "Linking Obesity and Cholesterol to Systemic Inflammation in Preeclampsia" is written carefully and in good language.
The work is a Review. A review article is an article that summarizes the current state of understanding on a presented topic. This type of work is expected to summarize the current state of knowledge, conduct discussions with contradictory or unclear results, and give the reader a comprehensive and systematic way with the accurate state of knowledge. The work of the Review type should also introduce an element of novelty, although it is secondary knowledge, the work should also bring a new quality, allowing to draw constructive conclusions. Meanwhile, the authors divided the work into several thematic chapters and presented a contribution of these factors in PE development. Quoting publications is made in the form of assertions without raising considerations about the problems solved by the authors. The work is a simple introduction to a topic rather than a Review.
The paper lacks a reliable presentation of the pathophysiology of PE as a background for further considerations. The presented information is very cursory; the paper does not provide any numerical values ​​to discuss, compare the results of various studies, analyze the statistical significance or conduct a meta-analysis type study. There are no tables that are characteristic of the work of the Review type and summarize the state of knowledge or numeric values presented in various publications. The paper presents only one elementary figure, which seems redundant, entitled "Proposed mechanism of maternal obesity and dyslipidemia promoting preeclampsia", while there is no figure that would collect and explain the inflammatory and cellular factors involved in the development of PE, including obesity and cholesterol, which are described in various chapters of the work.
The work does not make a clear distinction between in vitro studies (no such studies?), animal studies, and human studies and meta-analysis. As a result, the work is unclear; it is not clear what is the conclusion from basic studies and what the results are from the observation in patients.
The authors do not raise the safety of using the presented drugs during pregnancy - doubts are raised by statins or i-COX2, tadalafil, and sildenafil. What with other drugs? What are the benefits and what are the risks? If these are questionable issues, what is the state of the knowledge in the current publications? Especially since many of these drugs are contraindicated in pregnancy or even teratogenic. There is only one sentence about statins without any discussion.
The work also lacks a discussion that would summarize the current state of knowledge, show problems, ambiguities, issues requiring further research, and show what is proven. The conclusion sums up the work rather than introducing a new quality that results from the analysis of many publications.
Author Response
The work entitled "Linking Obesity and Cholesterol to Systemic Inflammation in Preeclampsia" is written carefully and in good language.
The work is a Review. A review article is an article that summarizes the current state of understanding on a presented topic. This type of work is expected to summarize the current state of knowledge, conduct discussions with contradictory or unclear results, and give the reader a comprehensive and systematic way with the accurate state of knowledge. The work of the Review type should also introduce an element of novelty, although it is secondary knowledge, the work should also bring a new quality, allowing to draw constructive conclusions. Meanwhile, the authors divided the work into several thematic chapters and presented a contribution of these factors in PE development. Quoting publications is made in the form of assertions without raising considerations about the problems solved by the authors. The work is a simple introduction to a topic rather than a Review.
We thank the reviewer for providing clarification. Because this review article is more an overview than a narrative review, we have changed the title to reflect the nature of this manuscript.
The paper lacks a reliable presentation of the pathophysiology of PE as a background for further considerations. The presented information is very cursory; the paper does not provide any numerical values ​​to discuss, compare the results of various studies, analyze the statistical significance or conduct a meta-analysis type study. There are no tables that are characteristic of the work of the Review type and summarize the state of knowledge or numeric values presented in various publications. The paper presents only one elementary figure, which seems redundant, entitled "Proposed mechanism of maternal obesity and dyslipidemia promoting preeclampsia", while there is no figure that would collect and explain the inflammatory and cellular factors involved in the development of PE, including obesity and cholesterol, which are described in various chapters of the work.
Thank you for the opportunity to clarify these multiple concerns. First, in an effort to address points brough up by both Reviewer 1 and 2, we have added detail and references providing mechanistic aspects of placental vasculature development and how it is perturbed in preeclampsia to the second paragraph.
Furthermore, we have added a table which summarizes human studies with and without nutrient interventions.
The figure is meant to be a graphical representation of the assimilated literature presented in this overview of maternal obesity, specifically dyslipidemia, and pregnancy outcomes. We have revised the title to be clearer of the purpose of the figure provided.
The work does not make a clear distinction between in vitro studies (no such studies?), animal studies, and human studies and meta-analysis. As a result, the work is unclear; it is not clear what is the conclusion from basic studies and what the results are from the observation in patients.
The authors thank the reviewer for the opportunity to clarify and expand on this point. No in vitro studies were included in this overview, and this has been expanded upon in text and as a table.
The authors do not raise the safety of using the presented drugs during pregnancy - doubts are raised by statins or i-COX2, tadalafil, and sildenafil. What with other drugs? What are the benefits and what are the risks? If these are questionable issues, what is the state of the knowledge in the current publications? Especially since many of these drugs are contraindicated in pregnancy or even teratogenic. There is only one sentence about statins without any discussion.
The authors respectfully agree with your comments on drug safety during pregnancy. More discussion with references have been provided on the therapeutics listed.
The work also lacks a discussion that would summarize the current state of knowledge, show problems, ambiguities, issues requiring further research, and show what is proven. The conclusion sums up the work rather than introducing a new quality that results from the analysis of many publications.
The authors have added substantially to the manuscript to this end and hope it is sufficient to clarify this point.
Round 2
Reviewer 2 Report
Good job. Please move animals, before human research. It will be more logical.